# Role of CARD Region of MDA5 Gene in Canine Influenza Virus Infection

**DOI:** 10.3390/v12030307

**Published:** 2020-03-12

**Authors:** Cheng Fu, Shaotang Ye, Yongbo Liu, Shoujun Li

**Affiliations:** 1College of Veterinary Medicine, South China Agricultural University, Guangzhou 510642, China; fucheng@zhku.edu.cn (C.F.); yeshaotang@stu.scau.edu.cn (S.Y.); 757259269@scau.stu.edu.cn (Y.L.); 2College of Animal Science & Technology, Zhongkai University of Agriculture and Engineering, Guangzhou 510225, China; 3Guangdong Provincial Key Laboratory of Prevention and Control for Severe Clinical Animal Diseases, Guangzhou 510642, China; 4Guangdong Technological Engineering Research Center for Pet, Guangzhou 510642, China

**Keywords:** MDA5, canine influenza virus, antiviral activity, innate immunity

## Abstract

MDA5 belongs to the RIG-I-like receptor family, which is involved in innate immunity. During viral infection, MDA5 generates an antiviral response by recognizing the ligand to activate interferon. However, the role and mechanism of MDA5 in canine influenza virus (CIV) infection are unclear. To understand the mechanism of canine MDA5-mediated innate immunity during CIV infection, we detected the distribution of MDA5 in beagles, and the structural prediction showed that MDA5 was mainly composed of a CARD domain, RD domain, and DExD/H helix structure. Moreover, we found that MDA5 inhibits CIV replication. Furthermore, in the dual luciferase assay, we revealed that the CARD region of MDA5 strongly activated the IFN-β promoter and mainly transmitted signals through the CARD region. Overexpression of the CARD region of MDA5 revealed that the MDA5-mediated signaling pathway could transmit signals by activating the IRF3/NF-κB and IRF3 promoters, promoting the expression of antiviral proteins and cytokine release, thereby inhibiting CIV replication. Upon silencing of MDA5, cytokine production decreased, while the replication ability of CIV was increased. Thus, this study revealed a novel mechanism by which MDA5 mediated CIV infection and provided new avenues for the development of antiviral strategies.

## 1. Introduction

Canine influenza (CI) is a contact respiratory disease caused by the canine influenza virus (CIV), belonging to the Orthomyxovirus type A influenza virus family [1]. CIV mainly invades the host’s respiratory system, causing clinical symptoms such as fever, cough, sneezing, loss of appetite, and depression in dogs, which can lead to respiratory failure and even death in the host [2,3]. At present, the main epidemic strain of CI is H3N2, but different subtypes have been reported in virus-infected dogs, including H1N1 [4,5], H10N8 [6], avian H9N2 [7], H5N1 [8,9], and avian H9N2 containing the canine influenza PA gene [10]. CIV was first reported in 2004, and molecular and antigenic analyses found that its genome is closely related to the equine influenza genome [11]. Although dogs are the natural host of CIV, researchers have successfully infected pigs and mice with the CIV [12,13], and researchers also isolated equine flu from pig and donkey in China [14,15], which spread across the host due to the high variability of influenza viruses. Therefore, further studies of canine innate immune response are important to control the spread of the virus. 

Innate immunity, as the first line of defense against pathogenic microorganisms, is an innate defense function. When pathogenic microorganisms invade the body, innate immunity is activated within minutes. The host–virus interaction is a prerequisite for host resistance, which occurs through the pattern recognition receptor (PRR) of the host cell. Host PRR can recognize the viral component of the pathogen-associated molecular pattern (PAMP). However, viral PRR is mainly identified by the Toll-like receptor (TLRs), RIG-I-like receptor (RLRs), and NOD-like receptor (NLRs) of the host cell. Among them, RLR is a cytoplasmic RNA helicase belonging to the DExD/H-box protein family, which mainly recognizes the nucleic acid of the virus through three molecules: retinoic acid induced gene I (RIG-I), laboratory of genetics and physiology 2 (LGP2), and melanoma differentiation associated gene 5 (MDA5) [16,17].

MDA5 is mainly composed of an N-terminal caspase recruiting domain (CARD) region, intermediate DExD/H helicase domain, and C-terminal repressor domain (RD) region [18,19]. The RD region of RIG-I is active, while that of MDA5 is inactive. Thus, up-regulation of the RD region in MDA5 can prevent viral-induced interferon [17]. Compared to MDA5 and RIG-I, LGP2 lacks a CARD region [19]. The primary function of the CARD region is to transmit signals, such as those to activate MAPK, NF-κB, and IRFs. The RD region is capable of binding to the 5′ region of virus RNA ligand. Additionally, the DExD/H helicase domain has RNA helicase function and ATPase activity [20]. Although MDA5 and RIG-I have similar structures, they recognize different ligands [21]. RIG-I mainly recognizes less than 1000 bp dsRNA and 5′ triphosphorylated RNA, while MDA5 mainly recognizes dsRNAs greater than 1000 bp [22]. Further, LGP2 regulates the expression of RIG-I and MDA5, depending on the type of viral RNA [23,24,25]. When MDA5 or RIG-I is activated by the ligand (RNA viruses), the exposed CARD region transmits signals through the CARD-CARD (CARD of MDA5 and CARD of IPS1) interaction, activating IRF3/IRF7 and NF-κB, inducing cytokine expression, and establishing a natural immune response [26,27,28,29]. 

CIV can be rapidly transmitted in dogs, and it is easily accompanied by secondary infections, causing greater harm to animal health. Early studies have found that MDA5 deficiency has been shown to increase susceptibility to virus infection [30,31,32]. Encephalomyocarditis virus failed to induce type I IFNs in MDA5-deficient mice, which increase viral infection [33,34]. MDA5 has been analyzed in many species [35,36,37], but the role of MDA5 in CIV virus infection is unclear. Therefore, further studies of the innate immune response of dogs will be important for controlling virus infection. In this study, canine MDA5 was cloned, and the role of the predicted functional domain of MDA5 was investigated. In addition, the main MDA5-mediated signaling pathways were examined. Because overexpression of MDA5 has shown antiviral activity against CIV in MDCK cells while inhibition of MDA5 gene expression has shown the opposite effect. These results will help to further understand the role of MDA5 in the canine immune system and provide new avenues for the development of antiviral strategies.

## 2. Materials and Methods 

### 2.1. Ethics Statement

All procedures in the animal experiments were approved by the South China Agricultural University Experimental Animal Welfare Ethics Committee with a permit number SYXK (YUE) 2014-0136.

### 2.2. Virus and Cell

H3N2 CIV (A/canine/Guangdong/02/2011, C/GD/02) was isolated in 2011 from a pet dog. Viruses were purified and propagated in the allantoic cavity of 9 to 11-day-old specific pathogen-free chicken eggs at 37 °C for 72 h. Following, the viruses were stored at −80 °C. Viral titers were determined by 50% egg-infectious dose (EID50) as calculated by the Reed Muench method [38].

Madin-Darby canine kidney (MDCK) cells were obtained from Shanghai Cell Bank, Type Culture Collection Committee, Chinese Academy of Sciences, China. Cells were cultured in Dulbecco’s modified Eagle medium (Gibco, Grand Island, NY, USA) with 10% fetal bovine serum (Biological Industries, Israel) at 37 °C and 5% CO_2_. 

### 2.3. Cloning and Bioinformatics Analysis of the Canine MDA5 Gene

Total RNA was extracted from the spleen using Trizol reagent (Takara, Otsu, Japan) according to the manufacturer’s instruction, and cDNA was reverse-transcribed using the HiScriptRII One Step RT-PCR kit (Vazyme, Nanjing, China). Based on the NCBI data for canine MDA5 (GenBank accession no. XP_545493), the open reading frame of MDA5 was selected, and the primers were designed to be amplified by Primer 5.0 software. PCR was performed with Phanta Max Super-Fidelity DNA Polymerase (Vazyme). The coding region gene of MDA5 was cloned into the pMD18-T vector (Takara) and sequenced using the SMART program to predict the amino acid sequence of canine MDA5.

### 2.4. Construction of Plasmids

The N-terminal CARD, the N-terminal CARD deleted, the C-terminal RD deleted, both C-terminal RD deleted and N-terminal CARD deleted, and the full-length MDA5 were inserted into the p3xFLAG with specific primers (Table 1). The constructed plasmids were named p3xFLAG-MDA5-CARD, p3xFLAG-MDA5-ΔCARD, p3xFLAG-MDA5-ΔRD, p3xFLAG-MDA5-ΔCARD+RD, and p3xFLAG-MDA5, respectively. 

### 2.5. Transfection

MDCK cells were grown to 60% confluence in 6-well cell culture plates, followed by transfection with siRNAs and plasmid using the Lipofectamine^®^ 3000 reagent (L3000015, ThermoFisher, Grand Island, NY, USA) according to the manufacturer’s instructions. Briefly, plasmids, or 50nM siRNA and 2 μL P3000, were diluted in 50 μL of serum-free OptiMEM medium. Further, 3 μL Lip3000 was diluted in 50 μL of serum-free OptiMEM medium. The dilutions were completely mixed and incubated at 25 °C for 15 min. The mixture was then pipetted into the medium and further cultured at 37 °C for 24 h. Following CIV infection, the cells were incubated in fresh medium at 37 °C for 48 h. The protein targeted for knockdown or overexpression was evaluated by western blotting.

### 2.6. Indirect Immunofluorescence Analysis

To determine the subcellular localization of the canine MDA5 gene, MDCK cells were evenly spread in the confocal dish and transfected with p3xFLAG-MDA5 plasmid at 60% confluence. At 36 h post-transfection, cells were fixed with 4% paraformaldehyde for 15 min at 4 °C and permeabilized with 0.2% Triton X-100 for 15 min. After blocking cells for 10 min with QuickBlock™ Blocking Buffer (Beyotime, Shanghai, China), cells were incubated with primary monoclonal mouse anti-Flag antibody (Beyotime, 1:1000) overnight at 4 °C. Subsequently, the cells were stained with Alexa Fluor 488-labeled Goat Anti-Mouse IgG (H+L) (1:500, Abcam, UK) for 1 h at 37 °C. Nuclei were stained with DAPI (Invitrogen, Grand Island, CA, USA) for 10 min at room temperature, and the samples were visualized using a confocal laser scanning microscope (Leica, Germany).

### 2.7. Luciferase Assay

MDCK cells were cultured overnight in 24-well plates at 37 °C until the cells grew to approximately 70%. Based on previous studies [39], canine IFN-β promoter, IRF-3 response element, and NF-κB response element were constructed.

MDCK cells were transient transfected with various expression plasmids (p3xFLAG-MDA5-CARD, p3xFLAG-MDA5-ΔCARD, p3xFLAG-MDA5-ΔRD, p3xFLAG-MDA5-ΔCARD+RD, and p3xFLAG-MDA5), empty control plasmid or poly I:C with Luciferase-expressing plasmids (pGL3-IFN-β, pGL3-IRF3, and pGL3-NF-Κb) and pRL-TK using Lipofectamine 3000 (Invitrogen). After 36 h post-transfection, cells were lysed and collected to evaluate luciferase activities using the Dual-Luciferase Assay Kit (Promega, WI, USA), according to the manufacturer’s instructions. All luciferase reporter assays were repeated three times.

### 2.8. Real-Time qPCR

Total RNA was extracted from cells, and cDNA was obtained using a reverse transcription kit PrimeScript-RT-Master-Mix (Takara). The SYBR Premix Ex TaqTM kit (Vazyme) was used for quantitative RT-PCR, and the experiment was carried out using the LightCycler 480 (Roche). Each sample was repeated three times using the 2-Δ Δ Ct method to analysis the relative mRNA expression level [40]. Primers are shown in Table 2. Poly I:C was transfected as a positive control into MDCK cells using Lipofectamine 3000 (Invitrogen) [40,41].

### 2.9. Virus Growth Curve

p3xFLAG-MDA5, p3xFLAG-MDA5-CARD, and p3xFLAG were transiently transfected into MDCK cells using Lipofectamine 3000 (Invitrogen). After 24 h, cells were infected with H3N2 CIV for 1 h absorption. Supernatants were collected at 12 h, 24 h, 36 h, and 48 h. Following, the viral supernatant was measured by TCID50.

### 2.10. Western Blotting

Different doses of MDA5 were transfected into MDCK cells in 6-well plates. After 24 h, cells were infected with H3N2, and cell proteins were harvested 36 h later. First, the cells were washed three times with pre-cooled PBS. Subsequently, 200 μL of lysate containing PMSF (RIPA: PMSF = 100:1) was added to each well and lysed on ice for 15 min. After lysis, it was collected into an EP tube, and the collected liquid was placed in a pre-cooled 4 °C centrifuge and centrifuged at 12,000 rpm for 15 min. After centrifugation, the supernatant was transferred to a new EP tube, and the protein concentration was measured by the BCA method, followed by addition of 5×loading Buffer (40 μL). Further, the protein was denatured at 100 °C for 15 min. Subsequently, 30 μg proteins were extracted for SDS-PAGE, and the protein on the PAGE gel was transferred to the nitrocellulose membrane by a wet transfer method. Membranes were blocked with 5% skim milk powder for 1 h at 37 °C, then incubated with mouse anti-FLAG primary antibody (1:1000, Beyotime), polyclonal antibody rabbit anti-NP antibody (1:500, Sino Biological, China), or mouse anti-GAPDH antibody (1:1000, Goodhere Biotechnology, China) for 12 h at 4 °C. Next, membranes were incubated with secondary antibody with Alexa Fluor 680-labeled Goat Anti-Mouse/Rabbit IgG (H+L) (1:5000, Abcam) at 37 °C for 1 h. The nitrocellulose membrane was analyzed using the LI-COR near-infrared imaging instrument.

### 2.11. RNA Interference

Three siRNAs (siMDA5-1, siMDA5-2, and siMDA5-3) were designed for the canine MDA5 gene, and the sequences are shown in Table 3. SiRNA and si-negative were designed by RiboBio (Guangzhou, China). SiRNAs were transfected into MDCK cells using the Lipofectamine 3000 reagent, and the silencing efficiency was measured by RT-qPCR. Inflammatory cytokines and antiviral genes were detected by qPCR in the interference and negative control groups. After MDA5 silencing, the replication efficiency of the virus was detected by TCID50.

### 2.12. Animal Experiment

In brief, 40-day-old beagle dogs were raised at the Laboratory Animal Center of South China Agricultural University. Before being challenged with H3N2 CIV, the blood of the Beagle dog was used for a hemagglutination inhibition test, showing negative results for the next experiment. The virus strain was diluted to 106 EID50/mL for use, and the beagle dog was anesthetized with pentobarbital at 0.15 mL/kg and inoculation by intranasal [41]. A total dose of 1 mL was used for each dog in the challenge group, and 1 mL of PBS was added to the control group. On the third and seventh days after inoculation, three dogs in each group were euthanized with pentobarbital overdose, and the brain, heart, liver, spleen, lung, kidney, and trachea were collected. Total RNA was isolated using the Trizol reagent (Takara), and MDA5 mRNA expression was analyzed by RT-qPCR.

### 2.13. Statistical Analysis

Statistical analysis was performed with unpaired Student’s t-tests, as implemented in GraphPad Prism 5 software (mean ± SD; *n* = 3; * *p* < 0.05; ***p* < 0.01; *** *p* < 0.001; ^#^
*p* > 0.05).

## 3. Results

### 3.1. Characteristics of Canine MDA5

The MDA5 gene was cloned from the spleen of the dog, which consisted of 3089 bp and expressed 1029 amino acids. Sequence analysis revealed two CARD domains (located at 9–100 and 130–202), a restriction enzyme region (306–492) at the N-terminus, a conserved uncoiling structure (709–826), a RIG-I domain (908–1021) at the C-terminus. The protein sequence alignment showed close homology between dogs and *felis*, with the similarity rate reaching 84.44% (Figure 1A). To investigate the expression of MDA5 mRNA in normal tissues, total RNA was extracted from the heart, brain, liver, spleen, lung, kidney, and trachea. RT-qPCR was used to analyze MDA5 mRNA expression. As shown in Figure 1B, canine MDA5 was expressed in all tested tissues. It was strongly expressed in the liver, spleen, and lungs, which were 57.89-fold (*p* < 0.01), 52.35-fold (*p* < 0.01), and 34.27-fold (*p* < 0.01) times that of heart expression levels, respectively. To investigate whether MDA5 was involved in the antiviral response following CIV infection in dogs, MDA5 mRNA expression was measured in the spleen and lung on 0, 3, and 7 days after H3N2 CIV infection. As shown in Figure 1C, after H3N2 CIV inoculation, the gene expression levels of MDA5 in the spleen were significantly increased on the third day (17.89-fold, *p* < 0.01), followed by 22.47-fold on the seventh day (*p* < 0.01). Meanwhile, the expression level of MDA5 in the lung was significantly increased on the third day (11.13-fold, *p* < 0.01), and seventh day (4.22-fold, *p* < 0.05). To detect the position of MDA5 protein in the cells, we transfected p3xFLAG-MDA5 into MDCK cells, laser confocal detection found that the canine p3xFLAG-MDA5 fusion protein was mainly distributed in the cytoplasm, similar to localization in human cells [42] (Figure 1D).

### 3.2. Antiviral Effect of MDA5

MDA5 is capable of activating IFN-β and ISGs expression, which is important for combating viral infections. To evaluate the antiviral ability of MDA5, the virus titer was detected at different time points after canine influenza viral infection in MDCKs overexpressing MDA5. We first tested the relative expression of MDA5 mRNA to detect the effects of MDA5 interference and overexpression (Figure 2A). As shown in Figure 2 B, compared with the control group, the virus titer was low after p3xFLAG-MDA5 or p3xFLAG-MDA5-CARD transfection. In addition, by transiently transfecting different doses (2.5 μg, 5 μg, 10 μg) of p3xFLAG-MDA5, the antiviral effect of MDA5 was further verified by western blot. As shown in Figure 2D, as the dose of p3xFLAG-MDA5 increased, the expression of NP protein decreased, indicating that MDA5 had an inhibitory effect on viral proliferation. After MDA5 silencing, viral replication was further detected. As shown in Figure 2C, the replication titer of the influenza virus began to increase after 24 h and was significantly higher than that of the control group. This indicates that inhibition of the MDA5 gene can promote replication of the influenza virus, suggesting that canine MDA5 has an antiviral effect.

### 3.3. Functional Characteristics of Canine MDA5

It has been reported that MDA5 can active IFN-β in mammals. To investigate whether canine MDA5 could activate IFN-β expression, and to identify its key component involved in this process, mutant plasmids were constructed. As shown in Figure 3A, overexpression of p3xFLAG-MDA5 activated the IFN-β promoter (2.35-fold). Further, overexpression the canine MDA5 missing the RD domain (p3xFLAG-MDA5ΔRD) activated the IFN-β promoter more strongly than full-length MDA5 (4.56-fold, *p* < 0.05). Loss of the CARD domain (p3xFLAG-MDA5ΔCARD+ΔRD and p3xFLAG-MDA5ΔCARD) did not activate the promoter, whereas overexpression of p3xFLAG-MDA5-CARD strongly stimulated the IFN-β promoter (98.11-fold, *p* < 0.01). After stimulation of MDCK with the MDA5 agonist poly I:C, the promoter displayed a higher fluorescence value than the control (50.65-fold, *p* < 0.01). These results indicate that the RD domain has self-inhibitory ability, and the CARD is essential for activating IFN-β. To explore whether MDA5 was involved in the regulation of signaling pathways through NF-κB or IRF3, MDCK cells were transfected with the pGL3-IRF or pGL3-NF-κB plasmid and the functional domain of MDA5. As shown in Figure 3B,C, IRF3 and NF-κB were mainly activated by the CARD domain of MDA5. Specifically, the CARD region of MDA5 significantly activated the IRF3 promoter, and its fluorescence values were 151.71-fold and 16.63-fold, respectively (*p* < 0.01). Further, the ability to activate IRF3 was approximately 10-fold stronger than that of NF-κB. These results indicated that the MDA5-mediated signaling pathway mainly transmitted signals downstream by activating IRF3. Further, MDA5 transmitted signals mainly through the CARD region, which is capable of strongly stimulating the IFN-β promoter through IRF3 and NF-κB (Figure 3B,C). To verify that the expression differences of the above cytokines were not due to different plasmid expressions, we used western blot to detect the expression of five plasmids in MDCK cells, which showed no significant differences in protein expression (Figure 3D).

### 3.4. Overexpression of Canine MDA5 and CARD Induces Antiviral Activity and Cytokine Expression

To investigate antiviral gene and cytokine induction by MDA5, the expression levels of IPS1, Mx, OAS, STAT1, IFN-β, IL-1β, IL-2, IL-6, IL-8, LGP2, and RIG-I were analyzed by RT-qPCR. As shown in (Figure 4A), overexpression of the CARD region of MDA5 increased the mRNA expression level of IPS1 in the RIG-I pathway by 2.53-fold (*p* < 0.05). Additionally, activated IFN-stimulated genes (ISGs), OAS and STAT1, mRNA expression levels increased by 2.68-fold and 3.3-fold (*p* < 0.05), respectively. Mx was also effectively activated, with its expression level approximately 2.3 times of the vector group (*p* < 0.01). In contrast, the CARD region of MDA5 promoted expression of IL-2 (1.69-fold, *p* < 0.05) and IL-8 (9.92-fold, *p* < 0.01), but had no significant effect on the expression of the other four cytokines (Figure 4B). Moreover, LGP2 (9.85-fold, *p* < 0.01) and RIG-I (3.35-fold, *p* < 0.05) were activated by CARD (Figure 4C). The results indicated that after activation of MDA5, the RIG-I pathway and its downstream ISGs were activated by the CARD region to induce the production of antiviral proteins, thereby resisting viral invasion.

### 3.5. Canine MDA5 Knockdown Reduces Poly I:C-Stimulated IFN-β, Reduces Inflammatory Factors, and Decreases Antiviral Activity

Three specific siRNAs from different regions of MDA5 were synthesized to further explore the function of canine MDA5, and one was selected by RT-qPCR. As shown in (Figure 2A), siMDA5-2 significantly inhibited the expression of MDA5 (*p* < 0.05). The siMDA5-2 or siNC and pRL-TK were cotransfected with a pGL3-IFN-β promoter into MDCK cells. After 12 h, cells were treated with poly I:C and control, respectively. After MDA5 gene silencing, RT-qPCR was used to detect expression of the RIG-I pathway and its downstream ISGs gene (Figure 5). Expression of related genes was inhibited, IFN-β (5.2-fold, *p* < 0.05) and IPS1(7.6-fold, *p* < 0.05) were significantly decreased. Furthermore, Mx expression was significantly down-regulated, which was approximately 1/7 of the control group (*p* < 0.05). After MDA5 gene silencing, MDCK cells were stimulated with p3xFLAG-MDA5-CARD, and the expression level of the inhibited gene was up-regulated. These results indicated that knockdown of the MDA5 gene reduced inflammatory factor production and attenuated antiviral activity. Furthermore, canine MDA5 was found to be essential for activating antiviral responses, via the CARD region.

## 4. Discussion

The natural immune system is the first line of defense against host pathogen invasion. Upon detection of PAMPs by pattern recognition receptors (PRRs), host cells initiate a series of signaling cascades to induce expression of downstream antiviral genes, such as IFN and inflammatory cytokines, to inhibit replication of pathogenic microorganisms, eliminate pathogen-infected cells, and stimulate adaptive immune responses [43,44,45,46]. RLRs, a family of cytoplasmic RNA helicases, are key to the identification of viral PAMP and antiviral immune coordination processes. MDA5 receptors are important members of RLRs; therefore, the role of MDA5 in CIV infection is of great interest.

Canine MDA5 was successfully cloned, through amino acid prediction and bioinformatics analysis. It showed close homology with the cat family, suggesting that it may have similar functions to this species. Similar to other MDA5 gene structures [35,36,37,47], the canine MDA5 gene has three major structural regions, including two CARD regions at the N-terminus, a centrally located DExD/H supercoiled RNA-binding region, and an RD region at the C-terminus. Structural analysis showed that the homology of the CARD region was relatively low, and the homology between the DExD/H and RD region was relatively high of canine MDA5.

As MDA5 is an important protein in innate immunity, studying its tissue distribution aids a better understanding of its function. In this study, although MDA5 was expressed in all tissues with extensive expression profiles, the highest expression was found in the liver, followed by the spleen and lungs. Further, its expression in the brain and heart was relatively low. When dog infected with H3N2, the expression level of MDA5 in the spleen (17.89-fold, *p* < 0.01) was significantly increased on the third day and decreased on the seventh day (22.47-fold *p* < 0.01). However, Wei et al. found that when dog infected with H5N1, the expression of MDA5 in the spleen was significantly increased (*p* < 0.05) on the third day and decreased on the fifth day [48]. Therefore, MDA5 may exert an antiviral effect in the early stage after infection with attenuated H3N2, which remains longer after infection with virulent H5N1.

The functional characteristics of the domain predicted by canine MDA5 were analyzed in this study. CARD overexpression in canine MDA5 significantly induced and activated the IFN-β promoter, which is consistent with the function of MDA5 in other mammalian systems [35,36,47]. However, overexpression of canine p3xFLAG-MDA5-ΔCARD could not induce IFN-β promoter expression. These results suggested that the CARD domain of canine MDA5 has similar functions as other mammals and is involved in immune signaling [47,48,49]. When MDA5 is activated by the ligand, such as Poly IC, the exposed CARD region is recognized by RIPLET, TRIM25 [49], and a tetramer [20,50,51,52]. This tetramer recruits IPS1, which interacts with IPS1 via the CARD region of IPS1 to activate various signaling molecules and activate MAPKs, TANK-binding kinase 1, TBK1, and IkB kinase [49,53]. Further, activation of IRF3/7 and NF-κB transcription factors and phosphorylation of MAPKs activate AP-1 transcription factors [27]. These transcription factors then enter the nucleus and bind to and initiate expression of interferons and inflammatory factors [26].

In addition, overexpression of the canine MDA5 CARD region enhances the expression of antiviral proteins such as Mx, STAT1, and OAS. Canine MDA5 mediates signal transmission by activating antiviral and proinflammatory responses. Furthermore, overexpression of canine MDA5 lacking C-terminal RD showed a stronger ability to induce the IFN-β promoter than full-length canine MDA5. Studies have shown that the C-terminal RD region of human MDA5 neither activates nor inhibits promoter expression [25], suggesting that the RD region of MDA5 differs across species. In addition, CARD overexpression of canine MDA5 increased expression of the pattern recognition receptors RIG-I and LGP2. Studies have shown that IFN-α and IFN-β can activate MDA5, RIG-I, and TLR3 mRNA expression in human lung adenocarcinoma epithelial cells and human umbilical vein endothelial cells [54]. Further studies have shown that the MDA5 and RIG-I genes are activated by IFN-α [55,56]. Therefore, MDA5 increases the expression levels of RIG-I and LGP2 genes through the CARD region, due to up-regulated IFN-α and IFN-β stimulation.

Cytokine expression is stimulated by a variety of factors, of which MDA5 stimulated expression of a subset of cytokines. Numerous studies have shown that IFN-β and ISGs can reduce virus production when the host is infected with virus [57]. To investigate whether canine MDA5 could inhibit replication of the influenza virus, we studied the replication curve of CIV. We found that the replication titer of CIV was reduced at different time points, especially at 36 h with dog MDA5 overexpression (Figure 2A). Further, when MDA5 was silenced, the replication titer of CIV increased (Figure 2B). The results indicate that MDA5 inhibits CIV replication. Additionally, when MDCK cells were transfected with different doses of p3xFLAG-MDA5 and infected with CIV, the expression level of the viral protein NP decreased with the increased plasmid dose (Figure 2C). This reverse dependence may be due to MDA5-induced inflammatory responses that promote viral replication through the RIG-I pathway recruiting viral target cells.

In summary, we successfully cloned the canine MDA5 gene from dogs and studied the function of each region of canine MDA5. It was found that canine MDA5 exerts antiviral function mainly through the CARD region. In addition, canine MDA5 activated the RIG-I-like pathway and its downstream ISGs pathway, promoting the expression of antiviral proteins cytokine release, thereby inhibiting viral replication. This study will help to comprehensively understand the antiviral activity of canine MDA5 in innate immunity in dogs and provide a theoretical basis for future research.

## Figures and Tables

**Figure 1 viruses-12-00307-f001:**
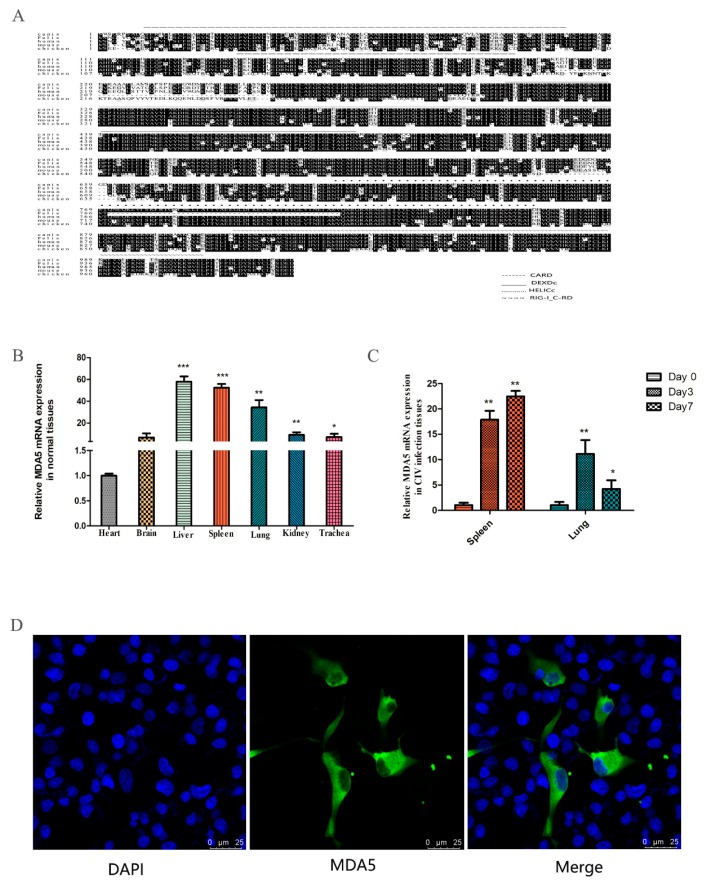
Characteristics of canine MDA5. (**A**) Amino acid alignment of *felis* (XP_011283648.1), *human* (NP_071451.2), *mouse* (NP_001157949.1), and *chicken* (NP_001180567.1). Alignment with clustal X program and edited with Boxshade. Dark shading indicates the same amino acid. The CARD, DEXDc, HELICc, and RIG-I_C-RD regions are indicated. (**B**) Tissue distribution of canine MDA5 in healthy dogs. The data are expressed as MDA5 mRNA relative expression, with the expression level of MDA5 in the heart serving as a control. Data were tested by one-way ANOVA with Fisher’s least significant difference (LSD) post hoc correction. * *p* < 0.05; ** *p* < 0.01; *** *p* < 0.001. Error bars indicate standard deviation. (**C**) Tissue distribution of MDA5 after infection with H3N2 CIV. Tissue distribution of MDA5 on the third (Day 3) and seventh (Day 7) day after infection with H3N2 CIV; the controls were inoculated with PBS. The relative MDA5 expression was calculated using the 2-ΔΔCt method. Data were tested by one-way ANOVA with Fisher’s least significant difference (LSD) post-hoc correction. * *p* < 0.05; ** *p* < 0.01; *** *p* < 0.001. Error bars indicate standard deviation. (**D**) Canine MDA5 subcellular localization. MDCK cells were transfected with p3xFLAG-MDA5 (2.5 μg/well) in a 14 mm-confocal dish. After 36 h, cells were fixed and stained. MDA5 was detected with mouse anti-flag antibody and Alexa Fluor 488-labeled Goat Anti-Mouse IgG (H+L). The result was obtained by laser confocal microscopy.

**Figure 2 viruses-12-00307-f002:**
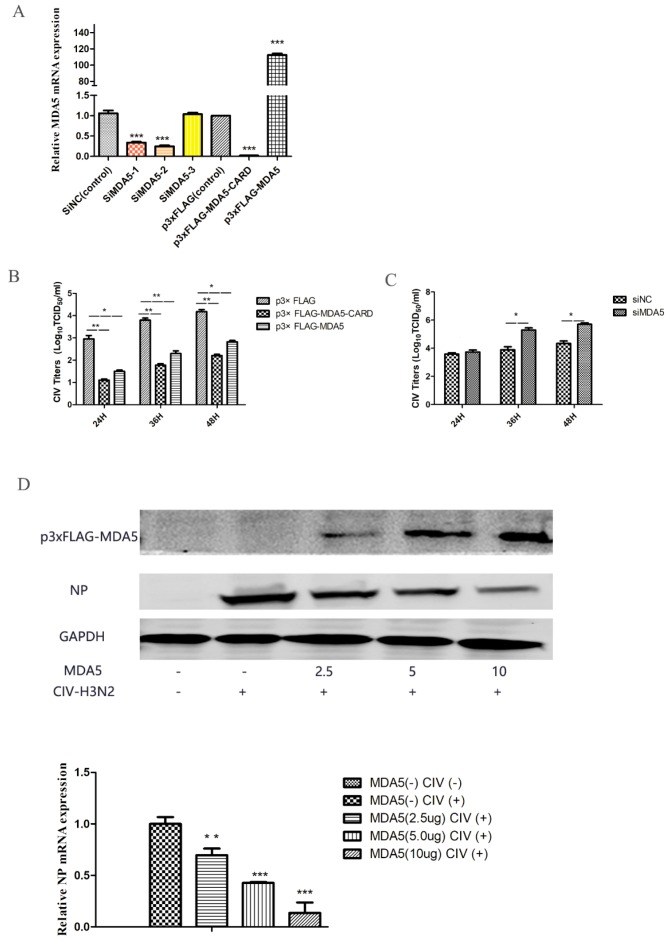
Effect of MDA5 and MDA5 CARD region on viral replication. (**A**) MDA5 mRNA was determined by RT-qPCR after transfection with siRNA and plasmid. Data were tested by one-way ANOVA with Fisher’s least significant difference (LSD) post- hoc correction. * *p* < 0.05; ** *p* < 0.01; *** *p* < 0.001. Error bars indicate standard deviation. (**B**) MDCK cells were transiently transfected with 2.5 μg/well p3xFLAG-MDA5-CARD, p3xFLAG-MDA5, and p3xFLAG in 6-well plates. After 24 h, MDCK cells were infected with 0.01 MOI doses of H3N2 CIV, and supernatants were collected at 24 h, 36 h, and 48 h after infection. The replication titer of the virus was detected by TCID50. (**C**) After transient transfection 2.5 μg/well of siMDA5-2 and siNC in 6-well plates, MDCK cells were infected with CIV 24 h later, and viral supernatants were collected at different time point (24 h, 36 h, 48 h) to detect viral replication by TCID50. (**D**) Different doses (2.5 μg, 5.0 μg, 10 μg) of p3xFLAG-MDA5 were transiently transfected. After 24 h, cells were infected with influenza virus at a dose of 0.01 MOI. After 36 h, the supernatant was collected to detect the expression of influenza virus NP protein and MDA5. Western blot analysis was used antibodies against Flag, GAPDH and viral NP. NP mRNA expression was tested by RT-qPCR. Data were tested by one-way ANOVA with Fisher’s least significant difference (LSD) post-hoc correction. * *p* < 0.05; ** *p* < 0.01; *** *p* < 0.001. Error bars indicate standard deviation.

**Figure 3 viruses-12-00307-f003:**
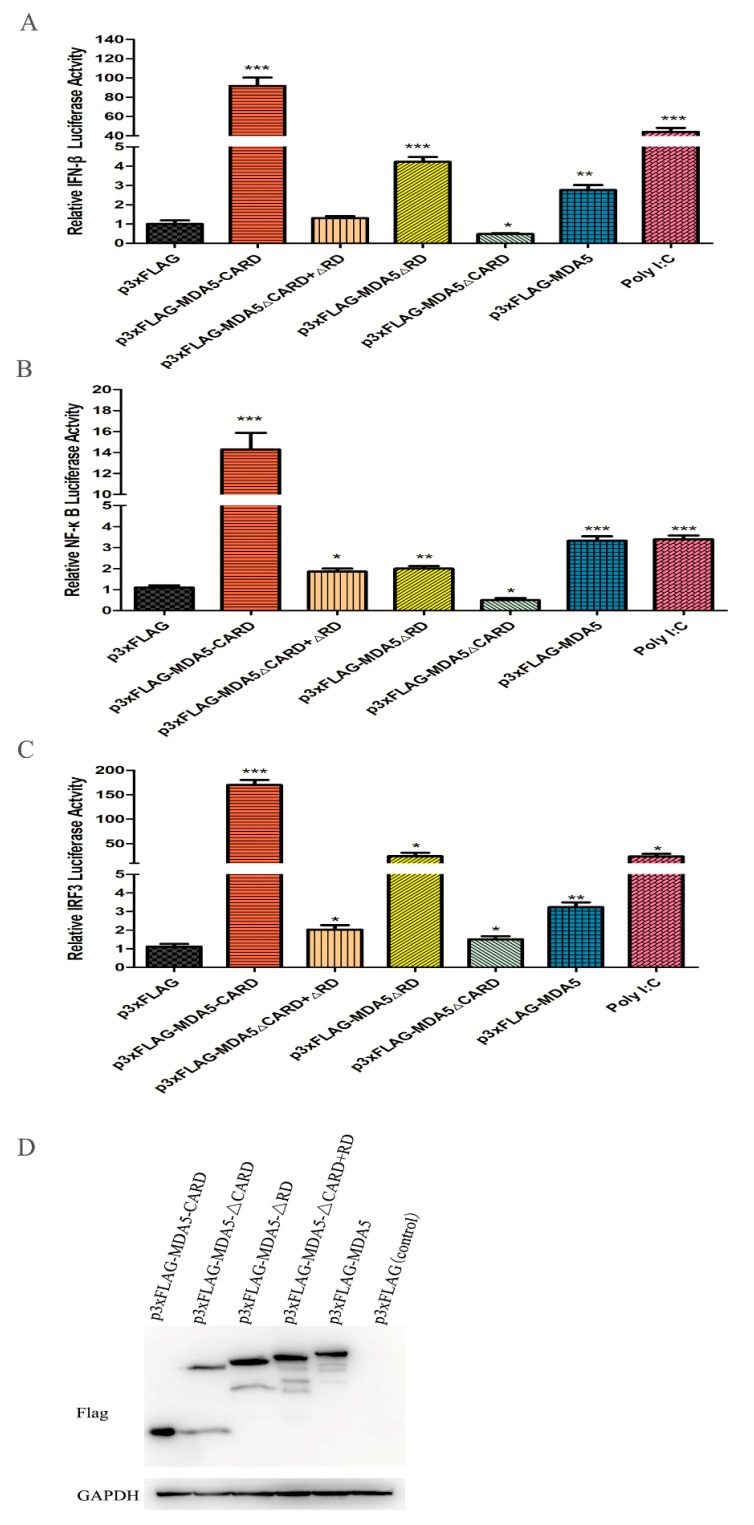
Effect of canine MDA5 domain on IFN-β/IRF3 and NF-κB promoter. p3xFLAG, p3xFLAG-MDA5-CARD, p3xFLAG-MDA5-ΔCARD, p3xFLAG-MDA5-ΔRD, p3xFLAG-MDA5-CARD+RD, p3xFLAG-MDA5 (500 ng/well), and poly I:C (500 ng/well) were transfected with pGL3-IFN-β (100 ng/well) (**A**), pGL3-NF-κB (100 ng/well) (**B**), pGL3-IRF3 (100 ng/well) (**C**), and pRL-TK (50 ng/well) in 24-well plates. Following, cells were harvested, and the relative fluorescein expression levels of IFN-β were measured after 36 h. Poly I:C represent a positive control. Data were tested by one-way ANOVA with Fisher’s least significant difference (LSD) post-hoc correction. * *p* < 0.05; ** *p* < 0.01; *** *p* < 0.001. Error bars indicate standard deviation. (**D**) The protein expression of five plasmids in MDCK cells was detected by western blot, and the GAPDH was used as a control.

**Figure 4 viruses-12-00307-f004:**
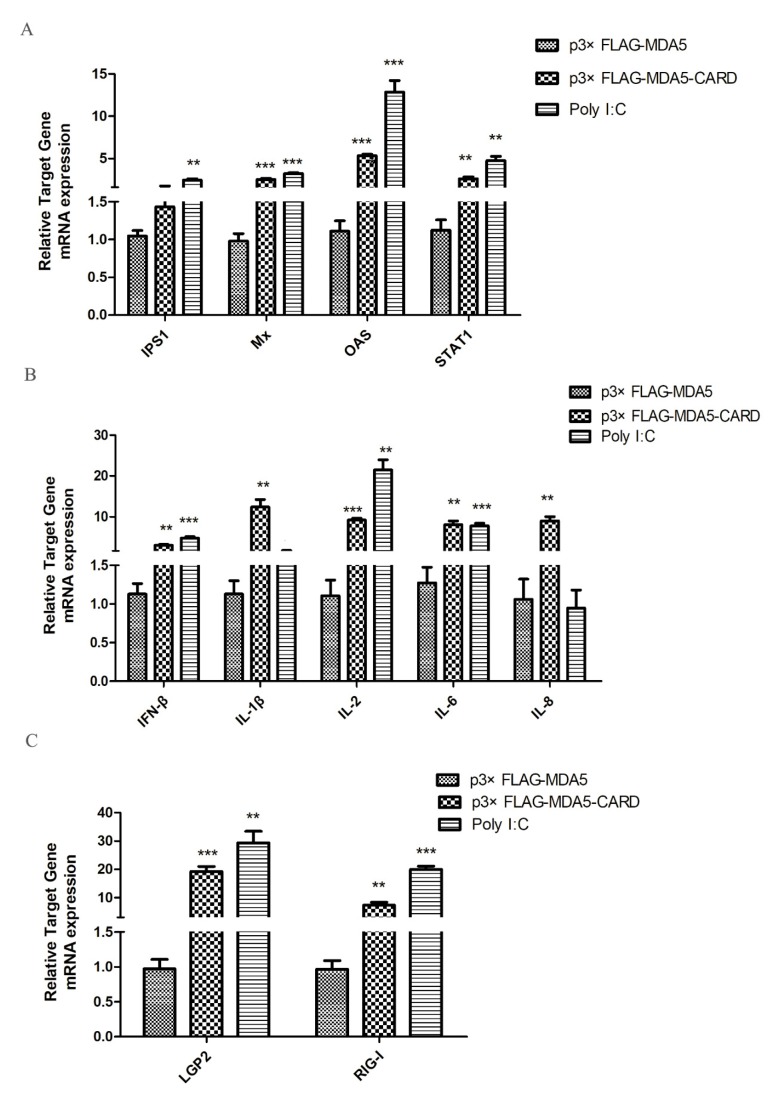
MDCK cellular response to p3xFLAG-CARD, p3xFLAG-MDA5, p3xFLAG, and poly I:C overexpression. The relative mRNA expression of (**A**) antiviral molecules (IPS1, Mx, OAS, STAT1), (**B**) proinflammatory cytokines (IFN-β, IL-1β, IL-2, IL-6, IL-8), and (**C**) RLR (LGP2 and RIG-I) was determined by RT-qPCR after 36 h transfection with p3xFLAG-CARD and p3xFLAG-MDA5 (2.5 μg/well) in 6-well plates. Poly I:C (2.5 μg/well) represents a positive control. Data were tested by one-way ANOVA with Fisher’s least significant difference (LSD) post-hoc correction. ** *p* < 0.01; *** *p* < 0.001. Error bars indicate standard deviation.

**Figure 5 viruses-12-00307-f005:**
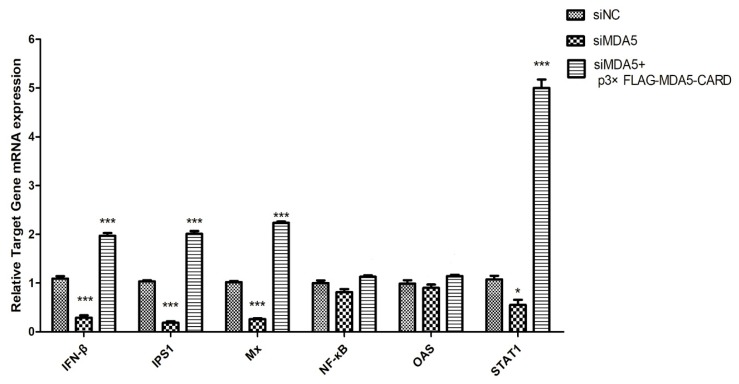
SiMDA5 attenuates antiviral activity. The effect of MDA5 after silencing against viral proteins. MDCK cells were transfected with siNC or siMDA5 (2.5 μg/well). After 36 h, MDA5 mRNA relative expression was measured. Further, siMDA5 was transfected for 12 h, and p3xFLAG-MDA5-CARD (2.5 μg/well) was overexpressed to measure the relative MDA5 mRNA expression. Data were tested by one-way ANOVA with Fisher’s least significant difference (LSD) post-hoc correction. * *p* < 0.05; *** *p* < 0.001. Error bars indicate standard deviation.

**Table 1 viruses-12-00307-t001:** Primers to identify each functional domain of MDA5.

Primer Name	Sequence of Oligonucleotides 5′~3′
MDA5△NRD-F	GAATTCAATGATGTGGAGCGGCCGC
MDA5△NRD-R	GGATCCTCAGTTTTCCTTGTAACACTTTGCAGC
MDA5△CARD-F	GAATTCAATGACTTGCTTTGAAAGCAAAGAAG
MDA5△CARD-R	GGATCCTCATCAATCCTCATCACTAAACAAAC
MDA5△NRD+CARD-F	GAATTCAATGACTTGCTTTGAAAGCAAAGAAG
MDA5△NRD+CARD-R	GGATCCTCAGTTTTCCTTGTAACACTTTGC
MDA5-CARD-F	GAATTCAATGATGTGGAGCGGCCGC
MDA5-CARD-R	GGATCCTCATGTGCCTGTTAGCTCTTGGAC
MDA5-F	GAATTCAATGATGTGGAGCGGCCGC
MDA5-R	GGATCCTCATCAATCCTCATCACTAAACAAAC

**Table 2 viruses-12-00307-t002:** Primers used for RT-qPCR analysis.

Primer Name	Forward Primer (5′~3′)	Reverse Primer (5′~3′)
IL-1B	TCAAGAACACAGTGGAATTTGAGTCTT	TCAGTTATATCCTGGCCACCTCTG
IL-6	TTCATTCCTTAGGATAGTGCTGAG	TCCTGAGGAGTGAAGATAACAATTT
IL-8	AAACACACTCCACACCTTTCCAT	GGCACACCTCATTTCCATTGAA
IL-2	AGTAACCTCAACTCCTGCCACAAT	TTGCTCCATCTGTTGCTCTGTTTC
RIG-I	CTCCAAGAAGAAGGCTGGTTC	AAGCAATCTATACTCCTCTAGACTTTC
LGP2	TCACTCCCTCCTACTCTGGCTC	TTTCGGATCACTTCTTGCTGGTCT
MX1	ATCACTGACTCGAATCCTGTACCC	GCCTACCTTCTCCTCATATTGGCT
OAS	CCAGGGTAACTCAGGAAGGAAAGT	CATCTCCATCAAACACGGGCTG
STA1	TTGACAGCAAAGTGAGAAACGTGA	ATTGGCTTCATGTTCTCGGTTCTG
IFN-β	GAAATCACGCCAGTTCCAGAAG	TCTCATTCCATCCTGTTCTAGAGATATT
TRIM25	TGAAACACTATATCAGGCAGTCCC	AAATGTATGGGTTTGTGCGTGGAT
TNF	CCCTGGTACGAGCCCATCTAC	AATGATTCCAAAGTACACCTGCCC
IPS1	GACCACAAGATGTCCGCAAGC	GGCAAGCTGTCTCTGGTGGA

**Table 3 viruses-12-00307-t003:** siRNA sequence used in RNA interference assays.

Name	Sequence of Oligonucleotides (5′~3′)
siMDA5-1	GACTGAGAATTTATCACAA
siMDA5-2	GTAGTTTCAGAATCAGACA
siMDA5-3	GTCATCACACCAACAAAGA

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
