# Peer review of "Role of CARD Region of MDA5 Gene in Canine Influenza Virus Infection"

_viruses, 2020, doi:10.3390/v12030307_

Round 1

Reviewer 1 Report

Viruses: Role of CARD region of MDA5 gene in canine influenza virus infection.

Overall, this is an interesting paper with a lot of data on canine MDA5.  However, the basis for this research is questionable as to date, there have been no reliable/peer reviewed published reports of CIV transmission from canines to humans, as stated in line 70.  Also, what is the end goal in understanding the role of MDA5 in CIV? How do you anticipate using this data?

A few specifics:

Line 19. “a beagle”  Just one beagle?

Line 52 “of the virus”  Which virus?  Viruses in general?

Line 59 Which signals does CARD transmit?  Please give a very brief over view of CARD and the purpose of the RD region binding to the RNA ligand.

Line 65.  Activated by which ligand?  Viral ligand?

Line 67  CARD-CARD ligand.  Are there two CARD regions on MDA5?

Line 69-71.  Based on a pubmed search and the most recent data from the US CDC, this is false.  Per the CDC website “In general, canine influenza viruses are thought to pose a low threat to people. To date, there is no evidence of spread of canine influenza viruses from dogs to people and there has not been a single reported case of human infection with a canine influenza virus in the U.S. or worldwide.”

Line 73:  “MDA5…. Has yet to be identified in dogs”.  If that is correct then why did you do this study? In line 74 you state that you are going to study canine MDA5 that was cloned. How do you clone something that doesn’t exist?

Line 88: Why did you use MDCK cells vs a canine cell line that has more flu tropism, like canine lung cells?

Line 93. Why did you extract RNA from the spleen vs another tissue?

Line 102.  What is the N-terminal CARD.  Is this the Wild type?  How does this differ from the full-length MDA5?

Line 110: Throughout the methods section you mention transfection or transient transfection but don’t provide a method.  Please include details on the two transfection methods.

Line 124: Poly IC vs Ploy IC

Line 133: Provide a reference for the 2DD Ct method.

Line 146.  What is “it” in “after the sample  was collected?  Collected the lysate?  What is an EP tube?

Line 149 Please provide a reference for the BCA method.

Line 152. Please provide a reference for the wet transfer method

Line 155 Would suggest changing Following to “Next”

Line 161.  Is an si-negative the negative control?

Lines 125,162.  Why do you use Lipofectamine 2000 for one assay and Lipofectamine 3000 for another?

Line 167: Animal experiment: This section needs a large amount of work.  Did you use A beagle, as stated in line 168?  If not,  how many beagles were used?  What virus strain?  Where were they inoculated?  How did you choose the dose of inoculum?  Why were they sedated when inoculated? How were they housed?

Line 180:  What do you mean by “Subsection”?  Sounds incomplete.  Is this the results section?

Line 182.  Could you provide an illustration or cartoon of how this canine MDA5 looks? It would be great if the illustration/cartoon could be side by side of a human, feline, MDA5

Line 194 Why use H3N2 vs N3N8?  This data should be included in methods

Line 198.  The last two sentences seem out of place and part of a different study vs the live dog study you are describing.

Figure 1B/C.  Where are the other tissues in Graph 1C? You measure the amount of MDA5 in cells but how much virus was in tissue going out to day 7?  Did you notice a decrease in virus over time?  Increase in certain tissues over others? 

Figure 2.  Please remind the reader of the control eg 3xFLAG (control), siNC(control)

Line 241.  Why did you use 0.01 MOI?

Line 279.  Was this done in tissue?  Cell culture?

Line 288. Seems out of place. Should this be in results or discussion?

Line 307  Would suggest changing the wording of “What’s more”.  Perhaps ending the sentence after “decreased” and beginning the next sentence with “Additionally” or “Furthermore”

Line 334.  The presence of two CARD regions should be mentioned in the intro

Lines 335.  In the sentence beginning with “Structurally analysis…showed homology was low …. Homology was high”  Homology between what?  Canine variants of MDA5?  Cat MDA5?

Line 341  “when infected”  Cells?  Dogs?  What/who was infected?

Line 343.  What was the viral titer on day 7?

Line 344. Wei has found when infected.  Again, who or what was infected with H5N1.

Line 374.  “the virus”  Do you mean CIV?

Line 376.  When the CIV titer was reduced, did you notice any mutations in the virus or MDA5

Line 380.  MDA5 promotes viral replication?  Figure 2A seems to contradict this statement.

Reviewer 2 Report

In this manuscript by Fu et al., the authors have characterized the pathogen recognition receptor (PRR) MDA5 in dogs. Although MDA5 has been extensively studied in mammals, there is no current report on dog MDA5. The authors did a comprehensive analysis of dog MDA5, from expression level in various tissues, to determination of functional domains, determination of signaling pathways, and capacity to modulate the viral replication of the Canine Influenza Virus (CIV) both by overexpression and knock-down of the protein of interest.

The study is well conducted and results are clearly presented. This will be of interest to scientists studying PRRs, as well as those interested in the sensing of viruses in dog hosts and viral cross-species transmissions.

Main revisions:

- Figure 1A is too small.

- The MDA5 basal expression in MDCK cells is missing.

Related: The authors should provide the validation of the siRNA in Figure 2 by mRNA titration. This may be added as a panel to Figure 2 with MDA5 mRNA in MDCK cells, with siMDA5, and with MDA5 2.5µg.

The authors may also provide the virus titers (as in Fig 2A) corresponding to the increasing dose of MDA5 (as in Fig 2C).

- Figure 3: The authors should provide the associated Western-blot for protein expression of the different constructs.

In particular, the authors need to exclude that the big difference they see between MDA5-CARD and MDA5 is not due to different expression levels.

Along the same line, the discussion in lines 362-366 should be undertoned (“RD has self-inhibitory ability”) and/or should be further investigated by the authors to state such conclusion. It is interesting if indeed all constructs express equally and if this is different from other mammalian MDA5s. But mechanisms and determinants should be further studied if the conclusion is kept as is.

Out-of-curiosity (not an asked revision): Have the authors tested MDA5 from human (or other species) in the canine cells/in the same settings (Fig3)? How is that different?

Minors:

- At the end of the introduction, lines 74-77: the authors may also add that their study includes silencing of endogenous MDA5 and associated findings. This would better highlight the key findings of their work.

- Lines 199-200: Can the authors compare/comment on the MDA5 distribution in dog cells vs other mammalian cells?

- In all figures: the authors should take out the signs # in the graphics. These are not necessary and they are misleading.

Few English typos and sentences should be corrected (e.g. line 28: “mediates” is wrong, line 68: Take out “as loyal friends of humans”, line 222: “overexpressing” > overexpress, line 247+248: “active” > activate, line 257: “INF” > IFN, Fig3: “Lucigerase” > Luciferase, line 366: “is differs” > differs, sentences in lines 377-382 should be reviewed for wording, line 376: add “in the context of overexpression of dog MDA5 (Fig 2A)”).

Round 2

Reviewer 1 Report

The revised manuscript “Role of CARD region of MDA5 gene in canine influenza virus infection” by Fu et al has been improved considerably.  There are a few minor comments/edits listed below.

There are a few grammatical /spelling errors that should be corrected with general spell check (eg line 21 “inhibiteds” line 138 “transfectly” line 140 “ploy” among some others)

Lines 39-42. The suggestion that CIV is zoonotic has been modified but the authors still suggest a risk to humans.  At this time there has been no documentation that this has occurred and should not be included in the manuscript.  The data is important enough to publish without trying to infer that it is or may become zoonotic.   Since it is important to note that experimental CIV infection has resulted in clinical signs in pigs and mice, perhaps the sentence can end with “high variability of influenza viruses”  Also, why isn’t the association with equine flu included?  It should be noted that reference 12 does not refer to CIV but to equine flu in pigs and should be mentioned as well.

Please include a reference for the 2-DD Ct method or is this part of the SYBR kit?

Animal experiments

Where were the dogs inoculated?  Was it a subcutaneous inoculation?

Line 214:  infection vs inoculation?

Author Response

There are a few grammatical /spelling errors that should be corrected with general spell check (eg line 21 “inhibiteds” line 138 “transfectly” line 140 “ploy” among some others)

Thank you for pointing out the mistakes. We have changed “inhibiteds” into “inhibits” in line 22, changed “these” into “These” in line 81, changed “transfectly” into “transfected” in line 138, changed “ploy” into “poly” in line 140, changed “Resultss” into “Results” in line 199, changed “oligonucletides” into “oligonucleotides” in table 1, changed “protein was” into “proteins were” in line 167-168.

Lines 39-42. The suggestion that CIV is zoonotic has been modified but the authors still suggest a risk to humans.  At this time there has been no documentation that this has occurred and should not be included in the manuscript.  The data is important enough to publish without trying to infer that it is or may become zoonotic.   Since it is important to note that experimental CIV infection has resulted in clinical signs in pigs and mice, perhaps the sentence can end with “high variability of influenza viruses”  Also, why isn’t the association with equine flu included?  It should be noted that reference 12 does not refer to CIV but to equine flu in pigs and should be mentioned as well.

Thank you very much for your comments. We have changed the expression in line 39-42, and added the associated expression about equine flu in line 40-43. The expression was changed into “CIV was first reported in 2004, and molecular and antigenic analyses found that its genome is similar to the equine influenza genome [11]. Although dogs are the natural host of CIV, researchers have successfully infected pigs and mice with the CIV[12,13], and researchers also isolated equine flu from pig and donkey in China,which spread across the host due to the high variability of influenza viruses[14,15]”.

Please include a reference for the 2-DD Ct method or is this part of the SYBR kit?

Thank you very much for your comments. It is not the part of the SYBR kit, and we have added the reference [40] in line 150.

Animal experiments

Where were the dogs inoculated?  Was it a subcutaneous inoculation?

Thank you very much for your comments. We utilized intranasal inoculation. We have added the expression and the reference [41] in line 190.

Line 214:  infection vs inoculation?

Thank you very much for your comments. We have changed “infection” into “inoculation” in line 213.